# Assessing the Impact of Workforce Nutrition Programmes on Nutrition, Health and Business Outcomes: A Review of the Global Evidence and Future Research Agenda

**DOI:** 10.3390/ijerph20095733

**Published:** 2023-05-05

**Authors:** Christina Nyhus Dhillon, Flaminia Ortenzi

**Affiliations:** The Global Alliance for Improved Nutrition (GAIN), 1202 Geneva, Switzerland

**Keywords:** workforce nutrition, nutrition outcomes, health outcomes, business outcomes, healthy food, health checks, nutrition counselling, nutrition education

## Abstract

One in three people globally suffers from at least one form of malnutrition, leading to poor health outcomes and low productivity in the workplace. The workplace offers an important, relatively unexploited opportunity to address malnutrition in all its forms. This narrative literature review aims to understand the impact of workforce nutrition programmes on nutrition, health, and business outcomes, based on high-strength-of-evidence studies. We used PubMed as our primary research database, complemented by Google Scholar, to identify systematic reviews, meta-analyses, and randomised controlled trials published between January 2010 and October 2021. In total, 26 records were included. We found that comprehensive workforce nutrition programmes, including a variety of intervention areas, and/or programmes targeting high-risk categories of workers (overweight/obese or (pre-)diabetic) were more likely to be effective on nutrition, health, and business outcomes. Within comprehensive and targeted programmes, individualised counselling and worksite environmental modifications were often mentioned as the most effective components. However, a high degree of heterogeneity in outcome measures and programme designs made it difficult to draw strong conclusions on the impact of workforce nutrition interventions. Limited evidence was found on business outcomes, longer-term effects of interventions, and programme implementation in LMICs. Therefore, further research is needed to address these evidence gaps.

## 1. Introduction

One in three people globally suffers from at least one form of malnutrition. For example, micronutrient deficiencies affect two out of three women and overweight/obesity affects at least one in ten people globally [1,2]. It is estimated that diet-related diseases result in 11 million deaths annually and 255 million disability-adjusted life years worldwide from poor health outcomes, including anaemia, hypertension, heart disease, and diabetes, among others [2]. Consequently, malnutrition of all forms also brings significant losses in productivity and work potential, posing challenges to employers in high-, low- and middle-income countries (HICs and LMICs) alike [3]. Given that about 60% of the global population will spend one-third of their time at work during their adult life and employers have an incentive to maximise performance, the workplace offers an important, relatively unexploited opportunity to address malnutrition in all its forms [3,4,5].

Workers commonly consume at least one meal during their working hours, whether operating from a field, a machine, or a desk. The environment in which people work can influence their access to and choice of foods, making worksites important avenues to promote adequate nutrition and healthy eating. Workforce nutrition programmes are interventions offered to workers through workplace delivery structures that intend to contribute to the nutritional needs of the worker population, which can encompass direct employees, indirect workers (in supply chains), and workers’ household members. Workforce nutrition interventions apply to any worksite setting and can focus on addressing a wide range of nutritional challenges, from healthy eating broadly, to consequences of poor nutrition including overweight/obesity, non-communicable diseases (NCDs), anaemia (and/or other micronutrient deficiencies), and/or underweight. Furthermore, workforce nutrition interventions are often integrated into wider employee health and well-being programmes, aimed at improving the physical, mental and social health of workers [3,4].

Workforce nutrition programmes have been characterized using a ‘four-pillar’ framework, which includes the following areas [6]:*Access to healthy food* interventions, which consist of the employer increasing access to nutritious foods for free or at subsidised costs, and/or making changes to the workplace food environment (e.g., healthier canteen menus, healthier snacks and beverages in vending machines, more balanced portion sizes and meal composition).*Nutrition education* programmes that aim to change employees’ dietary and/or lifestyle behaviours by increasing their nutritional knowledge and health literacy. Examples of interventions are cooperative menu planning, cooking demonstrations, dissemination of educational materials, interactive information sessions/workshops, and interpersonal communication.*Nutrition-focused health checks (and counselling)*, which are periodic one-to-one consultations with a health or nutrition professional to assess and discuss the employee’s nutritional and health status. Health checks help employees gain a better understanding of their nutritional and health risk factors, for example, through cholesterol and blood-pressure screenings, or weight monitoring and classification (Body Mass Index, BMI). Follow-up counselling can be provided in addition to health checks, to advise employees on potential dietary and lifestyle changes.*Breastfeeding support* interventions, which include programmes and/or policies aiming to enable working mothers to breastfeed their child exclusively for 6 months (i.e., providing only breastmilk to a child as per World Health Organization (WHO) recommendations) and continually up to 2 years [7]. Examples of policies and interventions are respecting or exceeding national laws on the duration of paid maternity leave, providing breastfeeding rooms and onsite childcare (where relevant), breastfeeding or breastmilk expression breaks, and flexible work schedules for mothers. In addition, breastfeeding support programmes can include awareness-raising or nutrition education campaigns for mothers and co-workers on the importance and benefits of breastfeeding.

This framework has been adopted and/or recognised by the Workforce Nutrition Alliance, the Access to Nutrition Index and the World Benchmarking Alliance, as well as numerous commitment makers in the Nutrition for Growth Summit in December 2021.

This research aims to investigate the impact of workforce nutrition programmes on nutrition, health, and business outcomes using the four-pillar framework. Specific objectives include the following: (1) to examine the extent, nature, and distribution of available high-strength-of-evidence literature; (2) to summarise and disseminate research findings, to inform policy and programmes; and (3) to identify potential gaps in the existing literature to recommend a future research agenda. In addition, by categorising and assessing the available evidence using this framework, this literature review will contribute to improving the comparability of reporting on nutrition, health, and business outcomes of workforce nutrition interventions, enhancing the standardisation of methods and indicators for future programme evaluations and evidence reviews. To our knowledge, this is the first time that a review assesses the existing high-strength-of-evidence literature on workforce nutrition interventions using the ‘four-pillar’ framework.

## 2. Methods

For transparency purposes we want to clarify that this study is a narrative review, not a systematic review; however, we wanted our methodology to be as structured and well-documented as possible to allow for reproducibility. The primary data source used was PubMed. Language and time search limits were set to the English language and January 2010-October 2021, to cover at least 10 years of published literature while including the most recent evidence available at the time of the search. Moreover, filters were applied for the following publication types: systematic reviews, meta-analyses, and randomised controlled trials (RCTs).

Two key concepts were identified: (1) the workplace setting, and (2) nutrition, based on which a series of relevant search terms were chosen. Search terms related to the first key concept included the following: “workplace” OR “workforce” OR “worksite”. Search terms related to the second key concept included the following: “nutrition” OR “nutrition policy” OR “nutrition program(me)” OR “nutrition intervention”.

The following search strategy was developed: at least one search term related to the first key concept AND one search term related to the second key concept had to be included in the title and/or abstract of articles, resulting in a total of 36 identified studies. The full search string used (including language, time, and publication type filters) is reported below.

(((“workplace”[Title/Abstract] AND “nutrition”[Title/Abstract]) OR (“workplace”[Title/Abstract] AND “nutrition”[Title/Abstract] AND “program”[Title/Abstract]) OR (“workplace”[Title/Abstract] AND “nutrition”[Title/Abstract] AND “policy”[Title/Abstract]) OR (“workplace”[Title/Abstract] AND “nutrition”[Title/Abstract] AND “intervention”[Title/Abstract]) OR (“workforce”[Title/Abstract] AND “nutrition”[Title/Abstract]) OR (“workforce”[Title/Abstract] AND “nutrition”[Title/Abstract] AND “program”[Title/Abstract]) OR (“workforce”[Title/Abstract] AND “nutrition”[Title/Abstract] AND “policy”[Title/Abstract]) OR (“workforce”[Title/Abstract] AND “nutrition”[Title/Abstract] AND “intervention”[Title/Abstract]) OR (“worksite”[Title/Abstract] AND “nutrition”[Title/Abstract]) OR (“worksite”[Title/Abstract] AND “nutrition”[Title/Abstract] AND “program”[Title/Abstract]) OR (“worksite”[Title/Abstract] AND “nutrition”[Title/Abstract] AND “policy”[Title/Abstract]) OR (“worksite”[Title/Abstract] AND “nutrition”[Title/Abstract] AND “intervention”[Title/Abstract])) AND (“systematic review”[Filter] AND 2010/01/01:2021/10/31[Date - Publication]) AND (“meta analysis”[Publication Type] OR “randomized controlled trial”[Publication Type] OR “systematic review”[Filter]) AND 2010/01/01:2021/10/31[Date - Publication]) AND ((meta-analysis[Filter] OR randomizedcontrolledtrial[Filter] OR systematicreview[Filter]) AND (2010/1/1:2021/10/31[pdat])).

The PubMed search was complemented by a search on Google Scholar, which yielded 16 additional studies.

The study selection process consisted of two subsequent stages: (1) initial screening of articles’ titles and abstracts for potential inclusion; and (2) screening of full article texts to confirm their eligibility for review. This was based on the following criteria:Having a clear focus on workforce nutrition, either as a stand-alone subject or as a principal component of broader workforce health and wellness (for example, records focusing solely on workplace physical activity interventions were excluded, while articles focusing both on worksite nutrition and physical activity programmes were included).Peer-reviewed articles with high strength of evidence (i.e., systematic reviews, meta-analyses and RCTs) (Figure 1). The main rationale for this choice is that the body of literature on workforce nutrition is increasing rapidly. However, only a small subset of it presents high strength of evidence.Touching on one or more of the four pillars of workforce nutrition policies and programmes described in the Introduction.Discussing the effects of workforce nutrition policies and programmes on nutrition, health and/or business (financial) outcomes.

After completion of the study selection process, a total of 26 records were identified as meeting the above criteria and were included in the review. The different stages of the selection process are summarised in a flow diagram, which is presented in Figure 2.

Relevant data from included records were extracted and organised by using a Data Charting Form to report on the following: (1) article details and general information, including title, author(s) and year, type of publication, primary aim(s), country/countries, income level(s), and workforce nutrition pillar(s) addressed; and (2) key aspects of interest to this review, including outcome measures used, programme duration, key findings, conclusions, lessons learned, and recommendations. The same extraction framework was applied systematically to all articles reviewed, aiming for a uniform, standardised approach.

A narrative review of the included studies was conducted, aiming to provide both descriptive and analytical insights, identify gaps in the existing literature, and make recommendations for future workforce nutrition policies, programmes, and research.

## 3. Results

### 3.1. Descriptive (Numerical) Analysis of Included Records

As mentioned above, at the end of the study selection process, a total of 26 records were included for review. Most of them (21 records) were systematic reviews (six of which included a meta-analysis) and the remaining (five records) were RCTs. The systematic reviews assessed studies with diverse research designs, ranging from RCTs (or cluster RCTs) to controlled trials (CTs), quasi-experimental studies, randomised crossover studies, modelling studies, pre-post-test or post-test studies, cohort studies, and cross-sectional studies, among others. Most systematic reviews (17 out of 21, or 81%) included at least one RCT, with three of them including only RCTs and one including only RCTs and CTs. Although a few systematic reviews (3 out of 21, or 14%) reported that the quality of included studies was high (often expressed as a low risk of bias), most of them (18 out of 21 or 86%) mentioned that the quality of studies was overall low-to-moderate.

More than two-thirds of all reviewed publications (18 records or 69%) focused on high-income countries (HICs); five included both HICs and upper-middle-income countries (UMICs); two included both HICs and lower-middle-income countries (LMICs); and only one record focused solely on LMICs.

In terms of workforce nutrition pillars addressed, most records assessed interventions including health checks and counselling and nutrition education combined (eight records, or 31%) or comprehensive programmes comprising access to healthy food, health checks and counselling, and nutrition education altogether (eight records or 31%). As for the remaining records, four discussed stand-alone breastfeeding support programmes; two reported on stand-alone access to healthy food interventions; two on stand-alone nutrition education programmes; and two on interventions including access to healthy food and nutrition education combined. No records assessing stand-alone health checks and counselling programmes were found.

Regarding the types of outcomes measured, the vast majority of included studies (20 records, or 77%) assessed a variety of nutrition and/or health outcomes, while less than half (12 records) evaluated business outcomes, and only four measured breastfeeding-specific outcomes.

Table 1 summarises key findings from the descriptive analysis above.

### 3.2. Content (Thematic) Analysis of Included Records

The results from the content analysis have been organised against the four workforce nutrition pillars presented in the Introduction, either as stand-alone interventions or as comprehensive programmes including two or more pillars. For additional details on each included record, please refer to the Data Charting Form available in the Appendix A document.

#### 3.2.1. Records Focusing on Access to Healthy Food as a Stand-Alone Intervention (One Pillar)

The two records discussing access to healthy food interventions alone focused on HICs only. One of them was an RCT targeting urban, low-wage workers [9], and the other a systematic review of three RCTs targeting overweight/obese office workers [10]. All assessed interventions offered free, discounted or otherwise-incentivised nutritious foods and found positive effects on dietary outcomes but not on body weight [9,10]. Specifically, both records reported increased fruit intake, and one of them also reported higher vegetable intake [9,10]. Interestingly, the RCT conducted among low-wage workers showed an impact from employer-provided take-home food rations rather than from food provided at the worksite [9]. This approach, which also applied behavioural economics, resulted in greater consumption of home-cooked meals [9], thus potentially benefiting not only workers themselves but also their families, although outcomes on family members were not assessed.

#### 3.2.2. Records Focusing on Nutrition Education as a Stand-Alone Intervention (One Pillar)

Only two records—two RCTs covering both an LMIC and a HIC—focused on nutrition education interventions alone [11,12]. Overall, the evidence was mixed and inconclusive for the effects of stand-alone nutrition education programmes on health, nutrition, and business outcomes. In particular, one RCT targeting factory workers in Iran found positive initial results on knowledge and awareness levels, body weight, BMI, and biological indicators for diabetes and cardiovascular disease (CVD) but did not assess longer-term impact beyond three months [9]. The second RCT, which evaluated the effects of calorie labelling in six worksite cafeterias in the UK on workers’ caloric intake, found inconclusive results [11,12].

#### 3.2.3. Records Focusing on Health Checks and Counselling as a Stand-Alone Intervention (One Pillar)

No records were found as part of this review, which focused on health checks and counselling interventions alone. This component was always assessed as part of comprehensive programmes including two or more pillars.

#### 3.2.4. Records Focusing on Breastfeeding Support as Stand-Alone Interventions (One Pillar)

Four records—all systematic reviews—focused on workforce breastfeeding support programmes, two of which included some studies conducted in LMICs [13,14], while the other two only included studies in HICs [15,16]. Common across all breastfeeding support programmes were lactation rooms, breastfeeding breaks and some support services, and all interventions were implemented as part of worksite policies [13,14,15,16], in some cases to comply with national labour laws [15,16]. A consistently positive impact was found across all records on breastfeeding exclusivity and/or duration and—where business outcomes were assessed—job satisfaction [13,15] and maternal absenteeism from child illness (due to the health-protective effects conferred by breastmilk) [14].

In one record, a dose-dependent association was found between the number of breastfeeding support services provided and the rates of exclusive breastfeeding [16]. Interestingly, another record found that despite existing policies and programmes for breastfeeding support, depending on socioeconomic status and workplace settings, women were more or less aware of the programmes and subjectively perceived different levels of support [15]. For instance, women within the service industry in the United States reported lower levels of perceived breastfeeding support from their employers [15]. Finally, one record also reported decreased premature breastfeeding cessation upon returning to work as a result of breastfeeding support initiatives [13].

#### 3.2.5. Records including Health Checks and Counselling Combined with Nutrition Education Interventions (Two Pillars)

Eight records assessed workforce nutrition programmes that included health checks and counselling and nutrition education in combination [17,18,19,20,21,22,23,24], sometimes as part of broader employee health programmes comprising other (non-nutrition) components (e.g., physical activity, smoking cessation, alcohol consumption). All seven records which assessed nutrition and/or health outcomes (e.g., dietary behaviours, weight loss, BMI, blood pressure, cholesterol, A1C) included studies conducted in HICs [17,18,20,21,22,23,24], while only one record assessing business outcomes (absenteeism, productivity, and workability) included both studies in HICs and UMICs [19]. Two records focused on interventions aimed at preventing and managing diabetes among (pre-)diabetic workers [17,21], while all other records assessed non-targeted programmes (i.e., directed at the general worker population) mostly focused on overweight/obesity, weight management, and/or body composition. Overall, the evidence shows mixed results, with four records finding mostly positive outcomes [17,21,22,23] and four finding inconclusive results or no effect [18,19,20,24]. The mixed findings might be partially attributed to the wide variation in programme duration (from 1 day up to 8 years), methodology, and outcome measurements across studies. Of note, the two records assessing programmes targeting (pre-)diabetic workers (i.e., employee groups at increased health risk) reported positive results [17,21]. Personalised interventions comprising an individual counselling component also showed effectiveness [19,22,23]. Additional elements contributing to programme effectiveness were building healthy food environments, which enabled longer-term weight loss [22]; promoting self-efficacy, which improved participation levels [21]; allowing for direct interaction with health or nutrition professionals [23]; employing motivational theory [23]; and developing content relevant to workers’ specific needs [23].

In terms of business outcomes, results were inconclusive, as only about a quarter of analysed studies found positive effects on absenteeism, productivity, and workability. Among the few interventions which reported positive outcomes, about half were specifically targeting overweight/obese employees and those exhibiting high rates of sickness absences [19].

#### 3.2.6. Records including Access to Healthy Food Combined with Nutrition Education Interventions (Two Pillars)

Two systematic reviews analysed studies comprising both access to healthy food and nutrition education interventions. One record covered multiple HICs and a UMIC [25], while the other only included HICs [26]. Both records included studies of moderate to low quality, showing positive results on the intake of fruits and vegetables, although of a small magnitude and with limited evidence of sustained impact over the longer-term [25,26]. The authors of the two systematic reviews conclude that the effects of workforce nutrition interventions would be more evident if programmes adhered to higher quality standards and studies used more rigorous methodologies [25,26].

#### 3.2.7. Records including Access to Healthy Food, Health Checks and Counselling, and Nutrition Education Interventions Combined (Three Pillars)

Eight records assessed workforce nutrition programmes that included access to healthy food, health checks and counselling, and nutrition education in combination [27,28,29,30,31,32,33,34], sometimes embedded in broader employee health programmes including other (non-nutrition) elements (e.g., physical activity, smoking cessation, and alcohol consumption) [25,28,31,32].

Of the eight records, only two included studies conducted in LMICs or UMICs [33,34]. Four records assessed both nutrition and/or health and business outcomes [28,31,33,34]; two assessed only business outcomes [29,32]; and two only nutrition and/or health outcomes [27,30]. Nutrition and health outcomes measured included dietary behaviours (e.g., intakes of fat, fruits and vegetables, sweetened beverages, water, etc.), body weight, BMI, body composition, cholesterol, blood pressure, triglycerides, CVD risk, nutrition, and health knowledge. The business outcomes comprised absenteeism, presenteeism, productivity, compensation claims, cumulative days with fever, medical costs, and health expenses.

Most records showed positive effects on at least one nutrition and/or health outcome [28,30,31,34], especially on dietary behaviours, as well as at least some business outcomes [28,29,31,34]. However, two records found inconclusive or mixed results [32,33]. Overall, individualised counselling, worksite environmental modifications (e.g., menu changes, larger offering and/or strategic positioning of healthier alternatives, calorie labelling, price discounts on healthy foods, portion size control, etc.), and targeted interventions were often mentioned as elements leading to increased effectiveness of workplace nutrition programmes [27,29,34]. Of note, regarding programme duration, one record concluded that continuous programming is important to ensure sustained results, as positive outcomes obtained in the short term were lost after the programme’s cessation [31]. Finally, two records mentioned that available evidence is limited and of low quality [32,33]. For instance, one systematic review assessing business outcomes found positive financial returns in non-randomised studies only, whereas studies using stronger analytical methods (RCTs) reported negative financial returns [32].

## 4. Discussion

This review assessed the impact of workplace nutrition programmes on nutrition, health and business outcomes based on existing high-strength-of-evidence literature. Two-thirds of identified records included experiences exclusively from high-income countries, and most of them evaluated nutrition and/or health outcomes, while only about half assessed business outcomes. In addition, about three-fourths of records assessed programmes including two or more workforce nutrition pillars.

One of the key findings of this review is that two particular characteristics of workforce nutrition programmes were more likely to be effective on nutrition and/or health outcomes: (1) being comprehensive (i.e., including multiple pillars and/or forming part of larger employee well-being programmes) [28,29,30,31,34], and (2) targeting workers at increased health risk (e.g., overweight/obese or (pre-)diabetic employees) [17,19,21,27,29,34]. The second characteristic (targeted interventions for high-risk groups) also generally led to positive business outcomes [17,19,21,27,29,34]. Within comprehensive and targeted programmes, individualised counselling and worksite environmental modifications were often mentioned as the most effective components driving positive nutrition and health outcomes [27,29,34].

When looking at business outcomes, the evidence suggests that breastfeeding support programmes and health checks and counselling interventions showed the most positive results for employers [13,14,15,29,34]. Overall, there is a lack of evidence on the business case for access to healthy food and nutrition education (the remaining two pillars), although when included in comprehensive programmes, they seem to contribute to positive business outcomes [28,29,31,34]. This particular finding does not necessarily imply the ineffectiveness of stand-alone access to healthy food or nutrition education interventions on business outcomes, as it might be (at least partially) due to a variety of factors related to insufficient research and study quality, including methodological rigour, choice of outcomes, and time at which the outcomes were assessed.

Most studies assessed outcomes only in the short to medium term (up to 6 months), while less is known about longer-term effects. Thus, while there is evidence of the effectiveness of short-duration programmes [19,20,23], little can be said about whether the positive results on nutrition and/or health are maintained in the longer-term [25,26]. We suggest that longer-term programmes are needed to sustain impact over time. However, better strategies to ensure continued participation and active engagement are needed, as well as more and higher quality evidence on the effectiveness of long-duration interventions.

When assessing individual pillars one by one, specific elements seemed to be most effective and emerged across multiple studies, although the small number of records focusing on single-pillar programmes limited our ability to draw strong conclusions on their effectiveness as stand-alone interventions. As for access to healthy food interventions, free, discounted or otherwise subsidised provision was particularly effective. Environmental modifications towards healthier food environments in the workplace also had positive impacts. Of note, access to healthy food interventions always resulted in positive outcomes when combined with health checks and counselling components [28,30,31,34].

Regarding nutrition education, the overall evidence is scarce and shows mixed and inconclusive results for stand-alone nutrition education interventions [11,12]. Even when embedded in comprehensive programmes, studies did not highlight nutrition education as one of the most effective components. Indeed, most records stressed the importance of individualised counselling rather than mass campaigns or similar educational approaches. However, methodologies, outcome measurements, and programme duration for nutrition education varied greatly, which might have contributed to inconclusive results. Nutrition education as a supportive component to access healthy food or health checks and counselling interventions may be useful to inform workers on relevant nutrition messages.

As for health checks and counselling, no records analysed this pillar alone. When in combination with nutrition education, results were mixed overall, with the exception of targeted interventions for diabetes prevention and management, where mostly positive results were found [17,21].

Finally, with regard to breastfeeding support, the evidence for effectiveness is positive and consistent. All records found positive impacts on breastfeeding exclusivity and/or duration and—where business outcomes were assessed—job satisfaction and maternal absenteeism [13,14,15,16]. In addition, the more support services provided (e.g., breastfeeding rooms, breaks, pumps, counselling), the better the outcomes [13,16]. Of note, breastfeeding support programmes assessed were never embedded within broader workplace wellness interventions. In most cases, they were implemented to comply with national labour policy regulations. This may be a missed opportunity by employers to communicate the benefits of breastfeeding for working mothers and their children, to the advantage of both workers and employers themselves.

### 4.1. Identified Literature Gaps and Recommendations for Programmes and Research

We identified several gaps in the existing high-strength-of-evidence literature. First, there was a great deal of heterogeneity in the nutrition, health, and business outcome measures, as well as the ways in which programmes were designed, implemented, and evaluated. This heterogeneity made it difficult to draw conclusions on the impact of workforce nutrition programmes. Workforce nutrition is a fairly nascent theme in the literature; therefore, further research is needed with a consistent delineation of intervention types (e.g., based on the four-pillar or other existing frameworks), comparable methods, and outcome measures. Furthermore, increasingly standardised packages of evidence-based programme elements, linked to positive outcomes, will be helpful for generalising findings on impact.

There was also very limited evidence on business outcomes generated from workforce nutrition programmes, which was consistent across all pillars. This finding is aligned with the broader literature [3,35,36]. Therefore, additional studies assessing return on investment or productivity outcomes would be helpful to inform employers as they prioritise worker benefits. In particular, future research could investigate the most effective workforce nutrition programme components for positive business outcomes, if any.

In addition, few of the included records provided evidence of workforce nutrition programmes implemented in LMICs, or among low-wage workers (even in HICs), which is consistent with other findings [37]. Workers’ different income levels may affect programme effectiveness in various ways. For instance, two workers who are equally aware of the importance of consuming healthy foods may have very different abilities to afford such nutritious foods in practice, depending on their income levels. More rigorous studies in LMICs and low-wage worker settings are needed and would also serve to inform larger companies when considering extending programmes to their supply chains.

Moreover, most of the records reviewed only assessed programmes of short-term duration (from 1 day up to 3 months), with variable—if any—follow-up. Longer-term studies are needed to evaluate the sustainability of obtained results over time.

Finally, there were no studies including all four pillars, but given the benefits of comprehensive programmes, it is likely further synergies may be found. Company strategies that aim to improve broader worker health should consider comprehensive programmes comprising all four nutrition pillars, as well as more holistic well-being interventions (e.g., physical activity, mental health, smoking cessation).

### 4.2. Study Limitations

This literature review has some limitations. First, it does not include worksite physical activity interventions, as they are not part of the selected framework for workforce nutrition programmes. In addition, physical activity (unlike nutrition) may not be relevant to all workplace contexts, but rather to work settings where overweight/obesity and NCD prevalence are high. However, since nutrition and physical activity are correlated and often share the same outcome measures (e.g., body weight, BMI, body composition, biomedical indicators for NCD risk), future research could consider the contribution of physical activity interventions (where relevant) to nutrition, health, and business outcomes resulting from workforce nutrition programmes. Similarly, in workplace contexts where underweight and micronutrient deficiencies (e.g., iron deficiency anaemia) are the most common forms of malnutrition, the research could consider complementary interventions to improve food safety and water, sanitation, and hygiene conditions (WASH).

Second, the search strategy did not surface any literature related to more informal work sectors (e.g., farming or temporary labour), which are more likely to be covered under evidence from community nutrition and development programmes. Future research could expand on this literature review to include less formal work settings, aiming to provide useful insights to employers and other actors engaged in supply chains.

Third, most of the evidence on the impact of access to healthy food, nutrition education, and health checks and counselling interventions came from comprehensive employee health programmes. Therefore, it was difficult to assess the contribution of each individual pillar independently (except for breastfeeding support interventions). However, in many of the studies on comprehensive programmes, the authors highlighted the most effective components, and, overall, there was consensus across studies.

Finally, on the one hand, the choice to only include high-strength-of-evidence literature was meant to allow us to draw more reliable conclusions to inform research, policy and programmes, but on the other hand, it led to the exclusion of a large volume of low-strength-of-evidence literature from LMICs and low-wage worker settings, which—although less reliable—may provide useful insights for programmes and policy in those contexts.

## 5. Conclusions

Based on the evidence available, workforce nutrition programmes hold the potential to contribute to addressing poor nutrition and reducing the prevalence of diet-related NCDs [38], and even more so when nutrition interventions are part and parcel of broader employee health and well-being programmes. Workforce nutrition programmes may represent a useful approach to support the achievement of the Sustainable Development Goals (SDGs)—not just SDG2 (“zero hunger”), but more broadly all SDGs related to global health as well as economic and social development [39]. They can also represent an important contribution of the private sector, as the dominant employer of workforces globally, towards the SDGs. With increasing requirements on businesses towards Environmental, Social and Governance accountability and reporting, the establishment of workforce nutrition programmes can demonstrate businesses’ engagement towards “Social” sustainability, while possibly also resulting in positive business outcomes and thereby representing a win-win for employers and employees. Simultaneously, policymakers should consider increasing incentives and establishing minimum standards and requirements for businesses to include workforce nutrition programmes as part of labour benefits packages.

The recent development of the Workforce Nutrition Alliance and the commitments made by many large employers under the Nutrition for Growth Summit in 2021 suggest an increasing interest in nutrition programming for employees. As programmes expand, better quality evidence—on both nutrition and health and business outcomes—will be increasingly necessary. More rigorous studies and more homogenous approaches and measures will help assess the real impact of workforce nutrition programmes and strengthen the case for investments from both the public and private sectors.

## Figures and Tables

**Figure 1 ijerph-20-05733-f001:**
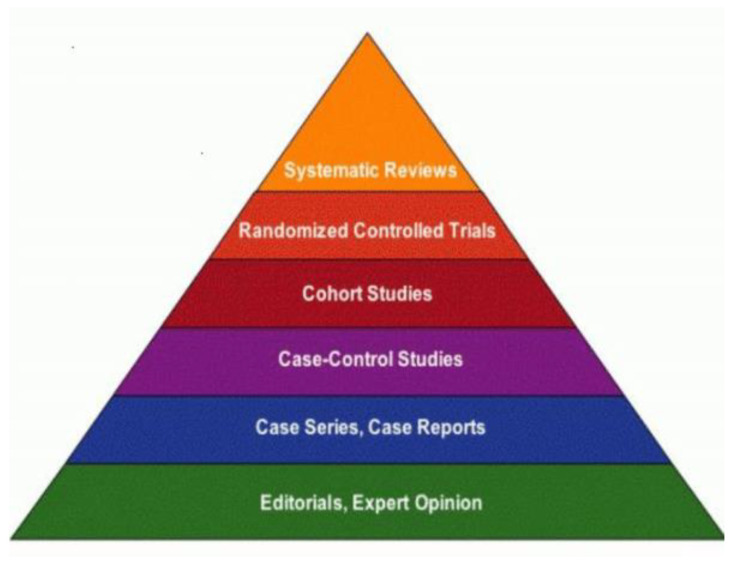
Strength of evidence pyramid [8].

**Figure 2 ijerph-20-05733-f002:**
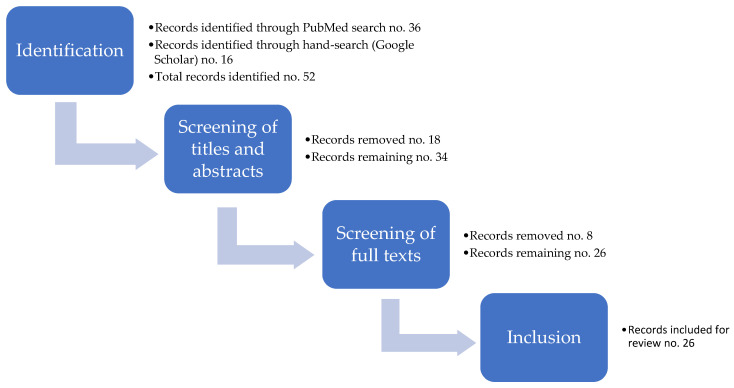
Flow diagram of the study selection process.

**Table 1 ijerph-20-05733-t001:** Summary of key findings from the descriptive analysis.

Aspect Considered	No. of Identified Records						
Type of publication	Systematic review				RCT		
	21				5		
Countries’ income level	HICs		HICs and UMICs		HICs and LMICs		LMICs
	18		5		2		1
Workforce nutrition pillar(s)	AHF ^a^	NE ^b^	HCC ^c^	BF ^d^	AHF + NE	HCC + NE	AHF + HCC + NE
	2	2	0	4	2	8	8
Outcomes measured	Nutrition and/or health		Breastfeeding		Business		
	20		4		12		

^a^ AHF: Access to healthy food; ^b^ NE: Nutrition education; ^c^ HCC: Health checks and counselling; ^d^ BF: Breastfeeding support.

## Data Availability

No new data were created or analysed in this study. Data sharing is not applicable to this article.

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
