# Peer review of "Assessing the Impact of Workforce Nutrition Programmes on Nutrition, Health and Business Outcomes: A Review of the Global Evidence and Future Research Agenda"

_ijerph, 2023, doi:10.3390/ijerph20095733_

Round 1
Reviewer 1 Report
The manuscript titled to Potential for Impact of Workforce Nutrition Programmes on Nutrition, 2 Health and Business Outcomes: a Review of the Global Evidence and 3 Future Research Agenda was critically examined.
The study creates the impression of a systematic review at first glance, but this study is a literature review. So I suggest editing some expressions. For example, this expression shown below can be deleted in the abstract and manuscript.
We conducted a systematic search on PubMed and a rapid hand-search on Google Scholar to identify systematic reviews, meta-analyses, and randomized controlled trials published between January 2010 and October 202.
Please check style of reference citations.
The study seems to be a systematic review with a few differences. It is not written in a reader friendly. Although the method has been written in detail, the findings are indeed complicated. It is not clear how the researchers came to a definite conclusion as a result of a literature review. Also, many limitations are mentioned, but this is a compilation. I think the article may mislead readers.
Reviewer 2 Report
This research undertake a systematic review to investigate the impact of workforce nutrition programmes on nutrition, health and business outcomes. The research also provides some suggestions for future research agenda, which is something that scholars are interested to know and see in scientific research.
Overall, the manuscript is interesting and adds to the academic body of literature and worth publication. There are some suggestions for improvement
I suggest you remove the term “potential for” from the title. Start the title with “The Impact of …………”
I think “breastfeeding support” could be removed from the key words.
Line 84 the literature review could be replaced by this research
Reasons for choosing the timeframe (January 2020-October 2021) in Line 98 should be provided.
I strongly suggest you add section about implications for business
The supplementary material link is not working. Please double check
Best wishes
Reviewer 3 Report
The review article titled "Potential for Impact of Workforce Nutrition Programmes on Nutrition, Health and Business Outcomes: a Review of the Global Evidence and Future Research Agenda" by Dhillon and Ortenzi highlights the impact of workforce nutrition programmes on nutrition, health, and business outcomes based on high-strength-of-evidence studies. The author mentions that a balanced diet and lack of micronutrients affect health in the long run. There is no doubt that a balanced diet is necessary to keep people healthy, and healthy people are more productive at their workplace. Therefore, it is very important for companies, institutes, and organizations to conduct periodic health awareness programs to keep people healthy. The article summarizes already published data between 2010 and 2021 from both high-, low-, and middle-income countries (HICs and LMICs).
However, there are a few points that need to be addressed by the authors:
1. The authors conducted a systematic search on PubMed and a rapid hand-search on Google Scholar to identify systematic reviews, meta-analyses, and randomized controlled trials published between January 2010 and October 2021. But I did notice one reference from 2005 in the References section. Could you please clarify it?
2. Individualized counseling and environmental modifications were often mentioned as the most effective components. Does people's income also play an important role in this?
3. The authors did not mention physical activity interventions on the worksite.
4. Which countries (HICs and LMICs) are more focused on comprehensive workforce nutrition programs or programs targeting high-risk categories of workers?
5. Some diseases are not related to diet but are caused by the work environment, such as hypertension. People who suffer a lot due to overload or not fulfilling the company's demands often fall into hypertension. Could you comment on this?
Round 2
Reviewer 1 Report
I still have a concern that this manuscript is a traditional review not systematic review. However, it contains method section and it still seems a systematic review.
Author Response
Thank you for this comment. We made it even clearer to the reader that this is a narrative review (not a systematic review), both in the abstract (see line 10 – “This narrative literature review…”) and in the methodology (see line 97 -
“For transparency purposes we want to clarify that this study is a narrative review, not a systematic review; however, we wanted our methodology to be as structured and well-documented as possible to allow for reproducibility.”).
We do not intend to remove the methods section because we consider this to be a strength of our manuscript. Unlike most narrative reviews, ours was conducted following a rigorous methodology which allows for reproducibility, and we feel this should be reflected in the paper. We hope that the additional edits we made are sufficient to clarify that this is a narrative review.
